# Neutralizing Activity against BQ.1.1, BN.1, and XBB.1 in Bivalent COVID-19 Vaccine Recipients: Comparison by the Types of Prior Infection and Vaccine Formulations

**DOI:** 10.3390/vaccines11081320

**Published:** 2023-08-04

**Authors:** Hak-Jun Hyun, Min-Joo Choi, Eliel Nham, Hye Seong, Jin-Gu Yoon, Ji-Yun Noh, Hee-Jin Cheong, Woo-Joo Kim, Sun-Kyung Yoon, Se-Jin Park, Won-Seok Gwak, June-Woo Lee, Byoung-Guk Kim, Joon-Young Song

**Affiliations:** 1Department of Infectious Diseases, Ajou University School of Medicine, Suwon 16499, Republic of Korea; hak-neck@hanmail.net; 2Department of Internal Medicine, International St. Mary’s Hospital, Catholic Kwandong University College of Medicine, Incheon 22711, Republic of Korea; cowgow@naver.com; 3Division of Infectious Diseases, Department of Internal Medicine, Korea University College of Medicine, Seoul 02841, Republic of Korea; neliel0106@gmail.com (E.N.); msmjoonhoo@gmail.com (H.S.); zephirisj9@gmail.com (J.-G.Y.); lundisoir@hanmail.net (J.-Y.N.); heejinmd@korea.ac.kr (H.-J.C.); wjkim@korea.ac.kr (W.-J.K.); 4Asia Pacific Influenza Institute, Korea University College of Medicine, Seoul 02841, Republic of Korea; 5Vaccine Innovation Center-KU Medicine (VIC-K), Seoul 02841, Republic of Korea; 6Division of Vaccine Clinical Research, Center for Vaccine Research National Institute of Infectious Diseases, Korea National Institute of Health, Korea Disease Control and Prevention Agency, Cheongju 28159, Republic of Korea; skyoon01@korea.kr (S.-K.Y.); sejin27@korea.kr (S.-J.P.); wsgwak@korea.kr (W.-S.G.); junewoo1213@korea.kr (J.-W.L.)

**Keywords:** SARS-CoV-2, bivalent vaccine, Omicron, neutralizing antibody, reactogenicity

## Abstract

Bivalent COVID-19 vaccines that contain BA.1 or BA.4/BA.5 have been introduced worldwide in response to pandemic waves of Omicron subvariants. This prospective cohort study was aimed to compare neutralizing antibodies (Nabs) against Omicron subvariants (BA.1, BA.5, BQ.1.1, BN.1, and XBB.1) before and 3–4 weeks after bivalent booster by the types of SARS-CoV-2 variants in prior infections and bivalent vaccine formulations. A total of 21 participants were included. Prior BA.1/BA.2-infected, and BA.5-infected participants showed significantly higher geometric mean titers of Nab compared to SARS-CoV-2-non-infected participants after bivalent booster (BA.1, 8156 vs. 4861 vs. 1636; BA.5, 6515 vs. 4861 vs. 915; BQ.1.1, 697 vs. 628 vs. 115; BN.1, 1402 vs. 1289 vs. 490; XBB.1, 434 vs. 355 vs. 144). When compared by bivalent vaccine formulations, Nab titers against studied subvariants after bivalent booster did not differ between BA.1 and BA.4/BA.5 bivalent vaccine (BA.1, 4886 vs. 5285; BA.5, 3320 vs. 4118; BQ.1.1, 311 vs. 572; BN.1, 1028 vs. 1095; XBB.1, 262 vs. 362). Both BA.1 and BA.4/BA.5 bivalent vaccines are immunogenic and provide enhanced neutralizing activities against Omicron subvariants. However, even after the bivalent booster, neutralizing activities against the later Omicron strains (BQ.1.1, BN.1, and XBB.1) would be insufficient to provide protection.

## 1. Introduction

Since the severe acute respiratory syndrome coronavirus-2 (SARS-CoV-2) Omicron variant BA.1 emerged as a variant of concern in November 2021, Omicron subvariants have continued to spread worldwide as of 2023 [1]. As of June 2023, approximately 768 million confirmed cases of coronavirus disease 2019 (COVID-19) were reported, and approximately 13 billion doses of COVID-19 vaccines were administered worldwide. In South Korea, about 32 million cases of COVID-19 were confirmed, and the majority of the cases (about 31 million) occurred in the era of Omicron and its subvariants [2].

Mass vaccination against SARS-CoV-2 has substantially reduced COVID-19 transmission and severity [3,4,5]. However, the effectiveness of COVID-19 vaccines targeting wild-type (WT) SARS-CoV-2 has declined in the Omicron era. A meta-analysis of COVID-19 vaccines showed that vaccine effectiveness declined to approximately 25% and 43% after the primary series and booster dose of the monovalent vaccine, respectively, in the Omicron era [3]. Previous studies showed that the third or fourth doses of COVID-19 monovalent vaccines did not sufficiently neutralize the BA.4/5 and newly prevalent Omicron subvariants [6,7,8,9,10]. The novel Pfizer–BioNTech bivalent vaccine containing antigens of WT and BA.4/BA.5 and the Moderna bivalent vaccine containing antigens of WT and BA.1 were developed and authorized for emergency use in September 2022 in response to the surge of Omicron subvariants [11]. These bivalent COVID-19 vaccines were introduced in October 2022 in South Korea, and the cumulative doses of bivalent COVID-19 vaccines are estimated to be 6.6 million in elderly individuals aged ≥60 years until late March of 2023 [12].

Since the latter half of 2022, the descendants of Omicron BA.2 have been persistently circulating. The representative descendants of BA.2 include BA.2.12, BA.2.75, BA.4, BA.5, and XBB. BQ.1.1 and BN.1, which are descendants of BA.5 and BA.2.75, respectively, have been prevalent globally after the extensive circulation of BA.5 [13]. In South Korea, BN.1 was the most prevalent strain in the first half of 2023, whereas the prevalence of XBB, a recombinant of two BA.2 descendants (BJ.1 and BM.1.1.1), is currently increasing (Figure 1) [14]. This trend differed from those of other countries, including the United States and Europe, where BQ.1.1 was the most prevalent subvariant and was replaced by XBB sublineages [15]. Decreased neutralizing activities against BQ.1.1 and XBB in monovalent and bivalent COVID-19 vaccine recipients have been reported; however, there is limited data on neutralizing activities against BN.1 [6,8,9,10]. The cross-reactive neutralizing activities elicited by bivalent vaccines would be helpful in predicting the effectiveness of these vaccines against Omicron subvariants and establishing future vaccination strategies.

This study aimed to determine the cross-reactive neutralizing activities of bivalent COVID-19 vaccines against Omicron subvariants, including currently emerging subvariants, and compare them by the formulations of bivalent vaccines and breakthrough infections.

## 2. Materials and Methods

### 2.1. Study Design and Procedure

This was a prospective cohort study conducted at the Korea University Guro Hospital in South Korea between October 2022 and March 2023. Eligible participants were immunocompetent adults (≥18 years) who had received at least three doses of the monovalent COVID-19 vaccine at least 6 months before the study enrollment and were scheduled to receive any of the bivalent BA.4/BA.5 or BA.1 vaccine. We collected data on the demographic characteristics (age, sex, height, and body weight), medical comorbidities (chronic heart disease, chronic lung disease, chronic liver disease, chronic renal disease, rheumatologic disease, hypertension, diabetes mellitus, and dyslipidemia), and COVID-19-related information (vaccination status and prior SARS-CoV-2 infection) of the participants at baseline. Blood samples were collected before (T0) and 3–4 weeks after (T1) administration of the bivalent vaccine. Prior SARS-CoV-2 infection was identified through self-reports and measurement of anti-nucleocapsid (N) antibodies in participants’ blood samples. Participants who were not diagnosed with COVID-19 and had negative results for anti-N antibodies were defined as the SARS-CoV-2 infection-naïve group (group 1). We defined the BA.1/BA.2-infected (group 2) and BA.5-infected (group 3) groups as participants diagnosed with COVID-19 in January–June and July–October 2022, respectively, based on the temporal prevalence of SARS-CoV-2 variants in South Korea. We measured immunoglobulin G (IgG) anti-receptor binding domain (RBD) antibodies before and after the bivalent booster and compared the neutralizing antibodies (Nabs) against WT and Omicron subvariants (BA.1, BA.5, BQ.1.1, BN.1, and XBB.1) among the subgroups. The subgroups were defined based on prior infection with SARS-CoV-2 variants and the formulation of the bivalent vaccine. Reactogenicity was estimated on day 7 after the administration of the bivalent vaccine using a questionnaire. The questionnaire included inquiries about injection site pain, erythema, fever, chills, myalgia, rash, headache, fatigue, dyspnea, arthralgia, vomiting, paralysis, and the need for antipyretic medications. It also captured information on the duration of these symptoms and whether outpatient clinic or emergency room visits were required.

### 2.2. Immunological Analysis

We measured IgG anti-RBD and anti-N antibodies and conducted neutralization assays on blood samples collected at T0 and T1. IgG anti-RBD antibodies were measured by Elecsys^®^ SARS-CoV-2 spike immunoassay (Roche Diagnostics, Basel, Switzerland) using Cobas 8000 (Roche, Basel, Switzerland) according to the manufacturer’s protocol. Anti-N antibodies were measured using the SARS-CoV-2 IgG assay (Abbott Laboratories, Chicago, IL, USA). A focus reduction neutralization test with 50% neutralization (FRNT50) was conducted to measure the neutralizing activity against SARS-CoV-2 variants. FRNT50 was conducted using WT (bCoV/Korea/KCDC03/2020 NCCP No. 43326), BA.1 (GRA: BA.1 NCCP No. 43408), BA.5 (GRA: BA.5 NCCP No. 43426), BQ.1.1 (GRA: BQ.1.1 NCCP No. 43427), BN.1 (GRA: BN.1 NCCP No. 43439), and XBB.1 (GRA: XBB.1 NCCP No. 43428) SARS-CoV-2. Serum samples were serially diluted from 1:20 to 1:43,740. The mixtures of serum dilution/virus were absorbed into Vero E6 cells and cultured in 96-well plates for 7.5 h. Immunostaining foci were visualized by sequentially incubating the mixture with SARS-CoV-2 Nucleoprotein (NP) rabbit Mab (Sino Biological, Beijing, China; 1:3000) and a secondary goat anti-rabbit IgG HRP-conjugated antibody (Bio-rad, Hercules, CA, USA; 1:2000). The Vero E6 cells were fixed with 4% paraformaldehyde and stained to visualize by TrueBlue Peroxidase substrate (Seracare, Milford, MA, USA). A reduction in foci count of 50% was calculated for the median-neutralizing titer (FRNT50) using a four-parameter logistic curve fit of SoftMax Pro GxP Software (Version 7.1.2.). The four-parameter logistic equation in the program is as follows: FRNT 50% = InterpX(Plot#1@Graph#1, 50).

### 2.3. Statistical Analysis

The geometric mean titers (GMTs) of anti-RBD IgG antibodies and Nab titers in each group were calculated by log transformation of the antibody titers, with 95% confidence intervals (CIs). Regarding intergroup comparisons, the Wilcoxon signed-rank test and Mann–Whitney U test were used for the paired and unpaired data, respectively. Statistical analyses were performed using Statistical Package for the Social Sciences version 20 (SPSS Inc., Chicago, IL, USA) or GraphPad Prism software (version 9.4; GraphPad Software, Inc., San Diego, CA, USA). Statistical significance was set at *p* < 0.05.

### 2.4. Ethics Statement

This study was reviewed and approved by the Institutional Review Board of Korea University Guro Hospital (approval no. 2021GR0099) and was conducted in accordance with the Declaration of Helsinki and Good Clinical Practice guidelines. Written informed consent was obtained from all participants at enrollment.

## 3. Results

### 3.1. Study Participants

A total of 21 participants (7 men and 14 women) were included in this study (Table 1). The median age and age ranges were 64 years (interquartile range [IQR]: 61–67 years) and 19–75 years, respectively. Nine (42.9%) participants had comorbidities. Nineteen participants (90.5%) received the fourth dose of the monovalent vaccine, and 2 participants (9.5%) received the third dose of the monovalent vaccine before the bivalent booster. The median interval between the last dose of the monovalent and bivalent vaccines was 217 days (IQR: 190–233 days), with a range of 183–280 days. Twelve participants (57.1%) received the BA.4/BA.5 bivalent vaccine, and 9 participants (42.9%) received the BA.1 bivalent vaccine. Sixteen participants (76.2%) had a prior SARS-CoV-2 infection. We classified 5 participants into the non-infected group (group 1), 10 into the BA.1/BA.2-infected group (group 2), and 6 into the BA.5-infected group (group 3) based on the timing of infection (Table 1). None of the participants were diagnosed with SARS-CoV-2 breakthrough infection during the follow-up period.

### 3.2. Immunological Analysis after Bivalent COVID-19 Vaccination

The GMTs of the anti-RBD IgG antibodies (U/mL) in the three groups at T0 and T1 are presented in Figure 2: group 1 (T0 vs. T1: 1650 vs. 29,303, *p* = 0.063), group 2 (9675 vs. 39,043, *p* = 0.002), and group 3 (20,233 vs. 44,762, *p* = 0.125). The GMT of the IgG anti-RBD antibodies was significantly higher in group 3 compared to groups 1 and 2 at T0 but did not differ among the three groups at T1 (Appendix A).

Neutralization assays against WT, BA.1, BA.5, BQ.1.1, BN.1, and XBB.1 SARS-CoV-2 in the three groups are presented in Figure 3 and Table 2. Nab titers against Omicron subvariants were negligibly low in group 1 at T0 (Table 2). Before the administration of bivalent vaccines (T0), Nab titers of group 3 were significantly higher against WT and all Omicron subvariants compared to group 1, but only for Omicron BA.5 (1849 vs. 672, *p* = 0.042), and XBB.1 (176 vs. 55, *p* = 0.007) when compared to group 2. Group 2 showed significantly higher Nab titers against the WT and most Omicron subvariants (except BA.5) than group 1 at T0. Participants with prior SARS-CoV-2 infection (groups 2 and 3) showed significantly higher Nab titers against all Omicron subvariants than non-infected participants (group 1) after bivalent vaccine immunization (T1). Nab titers against BQ.1.1 (GMT 115 [95% CIs, 73–183]) and XBB.1 (144 [79–261]) were lower than those for BA.1 and BA.5 in group 1 after the bivalent booster. Nab titers against all Omicron subvariants at T1 did not differ among groups 2 and 3. A comparison of the neutralizing activities within the groups is presented in Figure 3. Neutralizing activities against BA.1 and BA.5 did not differ among the three groups. Nab titers against BQ.1.1, BN.1, and XBB.1 were 8.0-fold, 1.9-fold, and 6.4-fold lower, respectively, than those against BA.5 in group 1. Nab titers against BQ.1.1, BN.1, and XBB.1 were significantly lower than those against BA.5 in groups 2 and 3, with fold-change values of 9.3, 4.6, and 15.0 for group 2 and 7.7, 3.8, and 13.7 for group 3, respectively. Nab titers against BN.1 were significantly higher than those against BQ.1.1 and XBB.1 in groups 2 (1402 vs. 697 vs. 434) and 3 (1289 vs. 628 vs. 355).

A comparison of the neutralization assay with the bivalent vaccine formulation is shown in Figure 4. Figure 4A shows a comparison of the neutralization assays in all the participants; no difference was observed between the two bivalent vaccine groups. We further compared the neutralization activity confined to participants with prior BA.1/BA.2 breakthrough infection (group 2) to reduce the confounding effect of the difference in SARS-CoV-2 variants in prior infections (Table 3). The neutralizing activities against the WT and Omicron subvariants did not differ between the two bivalent vaccine groups (Figure 4B and Table 3). The Nab titer against BQ.1.1 was higher in the BA.4/BA.5 bivalent vaccine group than that in the BA.1 bivalent vaccine group in group 2; however, the difference was not statistically significant (1275 vs. 466, *p* = 0.067).

### 3.3. Reactogenicity of the Bivalent COVID-19 Vaccine

A total of 20 adverse events were reported in 12 participants (7 participants after the BA.4/BA.5 bivalent booster and 5 participants after the BA.1 bivalent booster). The most common adverse event was injection site pain (9/21, 42.9%), followed by fatigue (23.8%), erythema (14.3%), chills (4.8%), headache (4.8%), and myalgia (4.8%) (Appendix A). No cases of fever, arthralgia, or rash were observed. All the symptoms resolved within 3 days of antipyretic medication administration. Six participants took antipyretic medications and did not visit the outpatient clinic or emergency room.

## 4. Discussions

This study demonstrated the cross-reactive neutralization activity against currently prevalent Omicron subvariants in participants who received the bivalent COVID-19 vaccine. We found that the bivalent booster increased the neutralizing activities against the WT and Omicron subvariants and was more prominent in participants with prior SARS-CoV-2 infection. The strain types of Omicron subvariants in previous infections and the bivalent vaccine formulations did not affect the neutralizing activities against Omicron subvariants. However, the neutralizing activities against BQ.1.1, BN.1, and XBB.1 were lower even after bivalent booster vaccination.

The emergence of new variants has provoked public health concerns owing to their transmissibility and virulence, which leads to severe illness. The transmissibility of SARS-CoV-2 variants is determined by their immune evasion properties and affinities to the angiotensin-converting enzyme 2 (ACE2) receptor, which is essential for the virus’s entry into human cells [16]. Nab titers are known to be a highly predictive factor of SARS-CoV-2 infection and can be used to predict the immune evasion properties of each variant virus [17]. In the present study, we found that the bivalent booster increased the cross-reactive neutralization activities against BQ.1.1, BN.1, and XBB.1. However, Nab titers after the bivalent booster were considerably lower in participants without previous infection with Omicron subvariants. Given the lower variant-specific Nab titers in these SARS-CoV-2 infection-naïve populations, even after a bivalent booster, an additional booster vaccination is recommended, particularly in the infection-naïve population at risk for severe diseases. In addition, the immune evasion properties of BQ.1.1 and XBB.1 from the bivalent vaccine-elicited Nabs and currently available monoclonal antibody agents (tixagevimab–cilgavimab; Evusheld^®^, AstraZeneca) indicate the need for an additional booster strategy for elderly individuals and immunocompromised populations [18].

In the present study, the neutralizing activities against the tested subvariants were not different between the BA.1 bivalent vaccine-boosted and BA.4/BA.5 bivalent vaccine-boosted groups. Previous exposure to Omicron subvariants, along with an Omicron-containing vaccine, can minimize differences in neutralizing activities against the bivalent vaccine-specific subvariants. In addition, this can be attributed to the suppressed immune response to variant epitopes after prime and booster vaccinations against WT and the subsequent dose of Omicron subvariants, which is referred to as immune imprinting [19]. Therefore, it is necessary to develop vaccines that can efficiently induce variant-specific antibody responses. Nevertheless, repeated administration of the monovalent COVID-19 vaccine or bivalent vaccine showed acceptable effectiveness for preventing severe disease [20]. In addition, the cross-reactive T-cell responses elicited by the bivalent vaccine can protect against severe infection by the currently prevalent Omicron subvariants [21]. Regarding safety, the reactogenicities after the bivalent vaccine were estimated to be lower than those after the monovalent vaccine. In the present study, 57.1% of the participants reported adverse events, and none reported fever. The frequency and intensity of solicited adverse events were lower than those after the primary second dose vaccination [22,23].

The most prevalent Omicron subvariant in South Korea as of April 2023 is BN.1, a descendant of BA.2.75. The BN.1 subvariant is not dominant worldwide; however, it has been prevalent in some countries, including India, Australia, and New Zealand [15,24]. Data on neutralization assays against BN.1 have been insufficient because the epidemic is limited to a few countries. In the present study, Nab titers against BN.1 after the bivalent booster were higher than those against BQ.1.1 and XBB.1 in SARS-CoV-2 infection-naïve participants (490 vs. 115 vs. 144). This trend was similar in the BA.1/BA.2-infected (1402 vs. 697 vs. 434) and BA.5-infected (1289 vs. 628 vs. 355) participants. Thus, although BQ.1.1 and BN.1 emerged at a similar time after the extensive circulation of BA.5 globally, BQ.1.1 outcompeted BN.1 in most countries after the introduction of bivalent COVID-19 vaccines. It is unclear why the BN.1 variant was more prevalent than the BQ.1 variant in some countries. A study demonstrated higher ACE2 binding affinity by BN.1 variant viruses compared with BA.4/BA.5 and BQ.1.1 variants [25]. Despite higher neutralizing activities against BN.1 over BQ.1.1, higher ACE2 binding affinity by the BN.1 variant may have increased its prevalence in South Korea.

XBB, a recombinant of two BA.2 descendants (BJ.1 and BM.1.1.1), and its sublineages (e.g., XBB.1 and XBB.1.5) have become predominant worldwide, replacing the preceding Omicron variants [15]. Many studies have reported high immune evasion properties of the XBB sublineages [6,8,9,10,21,26]. The degree of immune evasion properties of XBB.1 and XBB.1.5 is known to be similar; neutralizing activities against these subvariants after bivalent vaccination were considerably low [10,21,27]. In the present study, neutralizing activities against XBB.1 after the bivalent booster were the lowest among the tested subvariants. XBB.1.5 is known to possess a higher affinity for the ACE2 receptor than XBB.1, primarily due to the reversion of the F486S mutation in XBB.1 [27,28]. The lower neutralizing activities and higher transmissibility of XBB.1.5 may elucidate why XBB.1.5 is the most prevalent strain in South Korea and other countries. On 18 May 2023, the World Health Organization recommended a new formulation of a monovalent COVID-19 vaccine containing antigen of XBB.1 descendent lineage, excluding the wild-type strain to increase the new target antigen concentration and avoid immune imprinting [29]. The candidate antigens were XBB.1.5 or XBB.1.1.6, which have antigenic similarity with only two amino acid differences (E180V and T4278R) [29,30]. The introduction of this new COVID-19 vaccine might be helpful in mitigating the upcoming epidemic waves of XBB sublineages.

This study had some limitations. First, the sample size was small, although a statistically significant difference was found among the subgroups. Second, SARS-CoV-2 strains in prior infections were classified according to the timing of infection rather than by viral genetic sequencing. The strengths of this study include conducting neutralization assays for diverse SARS-CoV-2 subvariants, including BN.1, BQ.1.1, and XBB.1.

In conclusion, the BA.1 and BA.4/BA.5 bivalent COVID-19 vaccines are immunogenic and provide higher neutralizing activities against the Omicron subvariants. However, neutralizing activities against BN.1, BQ.1.1, and XBB.1 were insufficient even after bivalent booster vaccination. To anticipate for the upcoming XBB.1 sublineage epidemic and the emergence of new variants, viral surveillance should continue, and optimized vaccines should be developed to induce a stronger immune response against XBB sublineage viruses and avoid immune imprinting.

## Figures and Tables

**Figure 1 vaccines-11-01320-f001:**
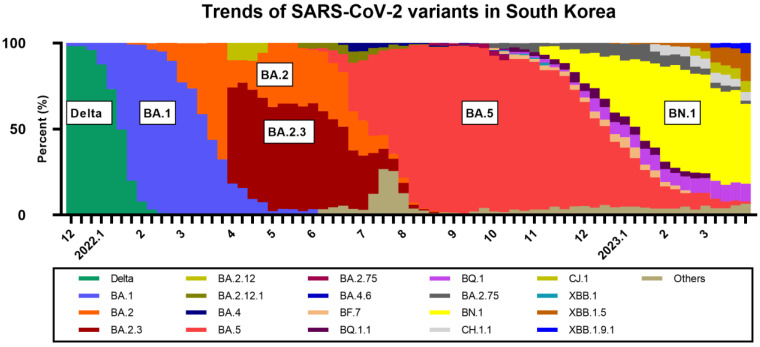
Trends of SARS-CoV-2 variants in South Korea from December 2021 to March 2023. Data are from weekly briefings of the Korea Disease Control and Prevention Agency (Available from: https://www.kdca.go.kr/contents.es?mid=a20107030000 (accessed on 10 April 2023).

**Figure 2 vaccines-11-01320-f002:**
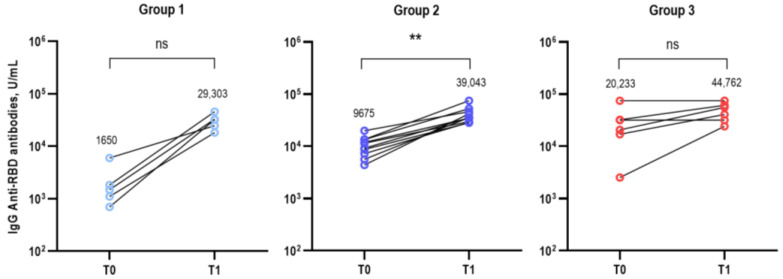
GMTs of the IgG anti-RBD antibodies in the SARS-CoV-2 infection-naïve (group 1), prior BA.1/BA.2-infected (group 2), and prior BA.5-infected (group 3) groups before (T0) and at 3–4 weeks (T1) after bivalent booster administration. Statistically significant *p*-values were marked with asterisks (** *p* < 0.01). Abbreviations: GMT, geometric mean titer; IgG, immunoglobulin G; RBD, receptor binding domain; ns, non-specific.

**Figure 3 vaccines-11-01320-f003:**
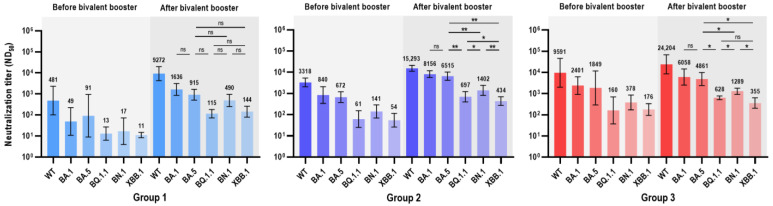
Comparison of ND_50_ against diverse Omicron subvariants in each group. GMTs of the neutralization titer (ND_50_) against WT and Omicron subvariants in SARS-CoV-2 infection-naïve (group 1), prior BA.1/BA.2-infected (group 2), and prior BA.5-infected (group 3) groups. Focus reduction neutralization test with 50% neutralization (FRNT_50_) was conducted to measure neutralizing activities against SARS-CoV-2 variants. Statistically significant *p*-values were marked with asterisks (* *p* < 0.05, ** *p* < 0.01). Abbreviations: GMT, geometric mean titer; ND_50_, 50% neutralization dose; FRNT_50_, focus reduction neutralization test with 50% neutralization; WT, wild type; ns, non-specific.

**Figure 4 vaccines-11-01320-f004:**
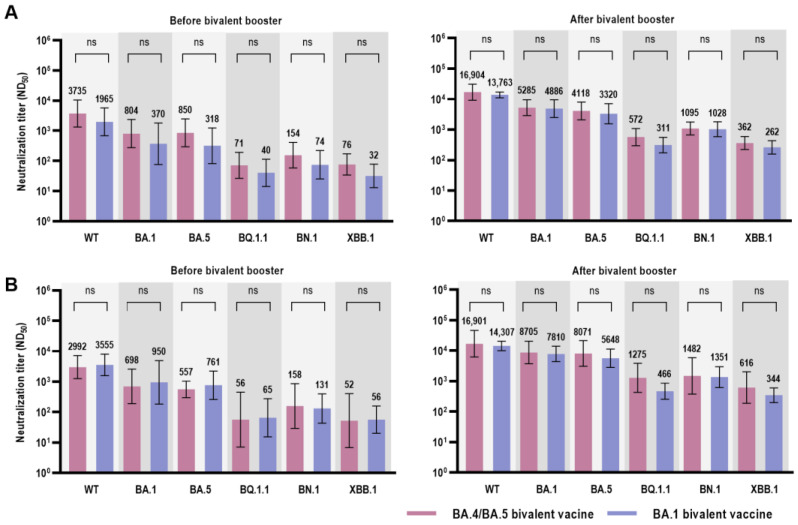
Comparison of the neutralization titer (ND_50_) against WT and Omicron subvariants between BA.1 bivalent vaccine and BA.4/BA.5 bivalent vaccine recipients. (**A**) Comparison of ND_50_ against WT and Omicron subvariants, including all participants. (**B**) Comparison of ND_50_ against WT and Omicron subvariants in the BA.1/BA.2-infected participants. Abbreviations: ND_50_, 50% neutralization dose; WT, wild type; ns, non-specific.

**Table 1 vaccines-11-01320-t001:** Baseline characteristics of study participants.

	Group 1	Group 2	Group 3
	SARS-CoV-2 Infection-Naïve (*n* = 5)	Prior BA.1/BA.2-Infected (*n* = 10)	Prior BA.5-Infected (*n* = 6)
Demographics			
Male, *n* (%)	2 (40.0)	2 (20.0)	3 (50.0)
Median age (IQR), y	65 (61–71)	63 (62–67)	64 (50–67)
Median BMI (IQR), kg/m^2^	21.8 (21.2–23.7)	25.4 (21.3–26.0)	24.8 (24.4–30.5)
Medical comorbidities, *n* (%)	2 (40.0)	5 (50.0)	2 (33.3)
Chronic heart disease	0 (0)	1 (10.0)	1 (16.7)
Chronic lung disease	0 (0)	0 (0)	0 (0)
Chronic liver disease	0 (0)	0 (0)	0 (0)
Chronic renal disease	0 (0)	0 (0)	0 (0)
Rheumatologic disease	0 (0)	1 (10.0)	0 (0)
Hypertension	1 (20.0)	1 (10.0)	1 (16.7)
Diabetes mellitus	2 (40.0)	2 (20.0)	2 (33.3)
Dyslipidemia	0 (0)	0 (0)	0 (0)
Previous vaccination, *n* (%)			
ChAd/ChAd/BNT/NVX	3 (60.0)	7 (70.0)	3 (50.0)
ChAd/ChAd/M/NVX	1 (20.0)	2 (20.0)	2 (33.3)
BNT/BNT/BNT/NVX	1 (20.0)	0 (0)	0 (0)
BNT/BNT/NVX	0 (0)	0 (0)	1 (16.7)
M/M/NVX	0 (0)	1 (10.0)	0 (0)
Median interval between previous dose and bivalent vaccination (IQR), days	190 (188–219)	216 (189–237)	230 (221–272)
Bivalent vaccine, *n* (%)			
BA.4/5	2 (40.0)	4 (40.0)	6 (100)
BA.1	3 (60.0)	6 (60.0)	0 (0)

Abbreviations: BMI, body mass index; IQR, interquartile range; ChAd, ChAdOx1 nCoV-19 vaccine (Oxford, AstraZeneca); BNT, BNT162b2 vaccine (Pfizer, BioNTech); M, mRNA-1273 (Moderna); NVX, NVX-CoV2373 vaccine (Novavax).

**Table 2 vaccines-11-01320-t002:** Intergroup comparison of neutralizing titers (FRNT_50_) by prior SARS-CoV-2 infection.

	Group 1	Group 2	Group 3	*p*-Value		
	SARS-CoV-2 Infection-Naïve (*n* = 5)	Prior BA.1/BA.2-Infected (*n* = 10)	Prior BA.5-Infected (*n* = 6)	Group 1 vs. Group 2	Group 2 vs. Group 3	Group 3 vs. Group 1
Before Bivalent COVID-19 Vaccine, GMT (95% CI)						
Wild type	481 (100–2300)	3318 (2061–5342)	9591 (1987–46,290)	0.028	0.056	0.009
BA.1	49 (11–224)	840 (339–2084)	2401 (920–6264)	0.003	0.093	0.004
BA.5	91 (9–939)	672 (376–1200)	1849 (290–11,799)	0.055	0.042	0.017
BQ.1.1	13 (6–27)	61 (25–152)	160 (37–693)	0.024	0.072	0.022
BN.1	17 (4–73)	141 (69–290)	378 (169–847)	0.007	0.056	0.009
XBB.1	11 (8–15)	55 (26–115)	176 (93–334)	0.016	0.007	0.004
After Bivalent COVID-19 Vaccine, GMT (95% CI)						
Wild type	9272 (4319–19,905)	15,293 (11,082–21,104)	24,204 (8558–68,450)	0.099	0.492	0.178
BA.1	1636 (850–3150)	8156 (5627–11,823)	6058 (2520–14,561)	0.001	0.264	0.009
BA.5	915 (506–1655)	6515 (4154–10,217)	4861 (2393–9875)	0.001	0.492	0.004
BQ.1.1	115 (73–183)	697 (397–1222)	628 (512–770)	0.001	0.713	0.004
BN.1	490 (251–958)	1402 (818–2404)	1289 (921–1804)	0.028	0.492	0.004
XBB.1	144 (79–261)	434 (271–695)	355 (201–626)	0.005	0.562	0.009

Abbreviations: FRNT_50_, focus reduction neutralization test with 50% neutralization; GMT, geometric mean titer; CI, confidence interval.

**Table 3 vaccines-11-01320-t003:** Neutralizing titers (FRNT_50_) after bivalent booster administration in participants with prior BA.1 infection.

	BNT162b2 (BA.4/BA.5 Bivalent)	mRNA-1273.214 (BA.1 Bivalent)	*p*-Value
Before Bivalent COVID-19 Vaccine, GMT (95% CI)			
Wild type	2992 (1252–7148)	3555 (1573–8035)	0.914
BA.1	698 (189–2579)	950 (184–4916)	0.914
BA.5	557 (298–1042)	761 (260–2222)	0.762
BQ.1.1	57 (7–450)	65 (15–274)	0.657
BN.1	158 (29–855)	131 (43–398)	0.762
XBB.1	52 (7–400)	56 (20–158)	0.814
After Bivalent COVID-19 Vaccine, GMT (95% CI)			
Wild type	16,901 (6204–46,047)	14,307 (10,047–20,372)	0.762
BA.1	8705 (3722–20,358)	7810 (4362–13,984)	0.914
BA.5	8071 (3060–21,287)	5648 (2842–11,224)	0.476
BQ.1.1	1275 (424–3837)	466 (254–855)	0.067
BN.1	1482 (373–5894)	1351 (613–2977)	0.762
XBB.1	616 (187–2023)	344 (198–599)	0.352

Abbreviations: FRNT_50_, focus reduction neutralization test with 50% neutralization; GMT, geometric mean titer; CI, confidence interval.

## Data Availability

No new data were created or analyzed in this study. Data sharing is not applicable to this article.

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
