# Peer review of "Neutralizing Activity against BQ.1.1, BN.1, and XBB.1 in Bivalent COVID-19 Vaccine Recipients: Comparison by the Types of Prior Infection and Vaccine Formulations"

_vaccines, 2023, doi:10.3390/vaccines11081320_

Round 1
Reviewer 1 Report
Comments to author:
Hakjun Hyun et al. reported the cross-reactive neutralizing activities of bivalent COVID-19 vaccines against Omicron subvariants, including BA.1, BA.5, BQ.1.1, BN.1, in 21 participants from South Korea. The authors found that the bivalent booster increased the neutralizing activities against the WT and Omicron subvariants and was more prominent in participants with prior SARS-CoV-2 infection. They also found that bivalent vaccine induced significantly lower neutralizing activities against BQ.1.1 and XBB.1 when compared to other Omicron subvariants. However, the low neutralization response against BQ.1.1, and XBB from mRNA bivalent booster have been well known and recently reported in NEJM (Meredith E. Davis-Gardner, et al. NEJM. 2023 Jan) and Nature Medicine (Chaitanya Kurhade et al. Nature Medicine. 2022 Nov). 2023 Feb, a study from Harvard and Beth Israel Deaconess Medical Center further reported substantial neutralization escape by SARS-CoV-2 Omicron Variants BQ.1.1 and XBB.1 after bivalent vaccination (Jessica Miller, et al. NEJM. 2023 Feb). Even though this is new neutralization data from South Korea participants against BQ.1.1 and XBB.1 and Omicron subvariants, the conclusion that manuscript got have no high impact on current mRNA vaccine community. The neutralization data of bivalent vaccine in human cohort has been fully reported and established. Based on these reasons, I will recommend this manuscript should be transferred to another journal or sub-journal.
English writing is logical and clear.
Reviewer 2 Report
This is a well planned and well executed study to evaluate the effect of bivalent COVID-19 Vaccines that contained BA.1 or BA4/BA5 omicron variants when tested as a booster to three groups of subjects. Group 1 was the control group that had not previously been infected, but was vaccinated and boosted. Groups 2 and Group 3 were previously infected respectively by omicron BA.1/BA.2 and BA.5 respectively. The results showed that the bivalent vaccines were immunogenic, but the levels of neutralising antibodies was insufficient to provide protection against the later strains of omicron virus that were tested. The main message of the paper is clearly stated in lines 222-224 , but not as clearly stated in the abstract. The main weakness of the paper is the limited number of study participants, which is acknowledged by the authors, with no justification. These reservations notwithstanding, the paper provides useful follow-up information about the use of bivalent COVID-vaccines. With minor editing to address the issues raised above, the paper can be accepted for publication in MDPI Vaccines.
Reviewer 3 Report
These are my comments:
Line 37-40, please also include data from the world.
Line 52-54, please also include data from the world.
Figure 1, the reference (web page or whatever) for this figure should also be included in the caption.
Line 95, please include what was evaluated in this questionnaire.
Line 115, please include the four-parameter logistic equation used.
Table 1, in 2.1, you did not state that information about comorbidities was to be gathered, please do so.
Table 1, use naïve in the second column (not naive), and throughout the document.
Line 156-157, are we sure that for group 2 the p value is 0.002. Hard to see this in Figure 2.
Table 2 and Figure 3A have the same information, use only the Table.
Table 2, some p values that are hard to believe, for example before booster, for XBB.1 you have for groups 2 and 3 values of 55(26-115) and 176(93-334), you report a p value of 0.007 when some of the data overlap. Please check all the p values in Table 2. Another example is after the booster, for the same strain, for group 1 you have 144(79-261) and for group 3 355(201-626), yet you reported a p value of 0.009 when some data clearly overlap.
Figure 3, figure 3B should be just Figure 3 (after removing part A). In Figure 3B, the ns or the asterisks are hard to see and again the significance or not is hard to believe. Check the p values. For example in cohort 2, when comparing BQ.1.1 (697) with BN.1 (1402) the error bars overlap each other, yet you have and asterisk which I assume is a p value less than 0.05. Also in Figure 3B, the caption should describe the single asterisk and the double asterisks.
Line 228, looking at Figure 4, I do not think there is significant differences among BQ.1.1, BN.1, and XBB.1, all showing the lowest responses.
Line 268-271, I do not believe there is statistically significant differences in the responses generated by BQ.1.1, BN.1, and XBB.1.
Line 301-302, I do not think that we can include BN.1 along with BA.1 and BA.4/BA.5 in terms of high neutralizing activity according to my previous comments.
Round 2
Reviewer 1 Report
Author properly made a response to the concern. There are some minor issues need to be addressed.
Pleses refer BN.1 genomic information or antigenic drift in introduction or results. This will help Vaccines readers to more understand why bivalent vaccines or which vaccine works better in against BN.1, not BQ.1 and XBB.1.
Line 72. 'March 2022' should be changed to 'March 2023'.
English writing looks good
Author Response
Thank you for the constructive comments of our manuscript. In response to your suggestions, we have revised our manuscript. We believe that by making these changes the manuscript is now much improved.
Q1. Pleses refer BN.1 genomic information or antigenic drift in introduction or results. This will help Vaccines readers to more understand why bivalent vaccines or which vaccine works better in against BN.1, not BQ.1 and XBB.1.
Answer)
Thank you for your valuable feedback. We agree with your suggestion that providing a clear phylogenetic lineage of the tested strains in our study can enhance readers’ understanding. As you recommended, we have provided a detailed description of the genomic information of the tested strains, and we have added a reference that readers can access further information about phylogenetic lineage. The revised manuscript is as follows.
Page 2, line 57–63: Since the latter half of 2022, the descendants of Omicron BA.2 have been persistently circulating. The representative descendants of BA.2 include BA.2.12, BA.2.75, BA.4, BA.5 and XBB. BQ.1.1 and BN.1 which are descendants of BA.5 and BA.2.75, respectively, have been prevalent globally after the extensive circulation of BA.5 [13]. In South Korea, BN.1 was the most prevalent strain in the first half of 2023, whereas the prevalence of XBB, a recombinant of two BA.2 descendants (BJ.1 and BM.1.1.1), is currently increasing (Figure 1) [14].
Page 12, line 287–290: Thus, although BQ.1.1 and BN.1 emerged at a similar time after the extensive circulation of BA.5 globally, BQ.1.1 outcompeted BN.1 in most countries after the introduction of bivalent COVID-19 vaccines.
Q2. Line 72. 'March 2022' should be changed to 'March 2023'.
Answer)
Thank you for your good comment. We have corrected the typo you mentioned.
Page 2, line 75: Figure 1. Trends of SARS-CoV-2 variants in South Korea from December 2021 to March 2023. Data are from weekly briefings of the Korea Disease Control and Prevention Agency (Available from: https://www.kdca.go.kr/contents.es?mid=a20107030000)
Reviewer 3 Report
The authors have addressed my previous comments.
Author Response
Thank you for the constructive comments of our manuscript. We greatly appreciate the improvement in our manuscript through your valuable comment.